# Multi-Camera Vessel-Speed Enforcement by Enhancing Detection and Re-Identification Techniques [note 1]

**DOI:** 10.3390/s21144659

**Published:** 2021-07-07

**Authors:** Matthijs H. Zwemer, Herman G. J. Groot, Rob G. J. Wijnhoven, Egor Bondarev, Peter H. N. de With

**Affiliations:** 1Department of Electrical Engineering, Eindhoven University of Technology, 5600 MB Eindhoven, The Netherlands; E.Bondarev@tue.nl (E.B.); P.H.N.de.With@tue.nl (P.H.N.d.W.); 2ViNotion B.V., 5641 JA Eindhoven, The Netherlands; rob.wijnhoven@vinotion.nl

**Keywords:** computer vision application, video surveillance, maritime traffic management, vessel detection, vessel re-identification

## Abstract

This paper presents a camera-based vessel-speed enforcement system based on two cameras. The proposed system detects and tracks vessels per camera view and employs a re-identification (re-ID) function for linking vessels between the two cameras based on multiple bounding-box images per vessel. Newly detected vessels in one camera (query) are compared to the gallery set of all vessels detected by the other camera. To train and evaluate the proposed detection and re-ID system, a new Vessel-reID dataset is introduced. This extensive dataset has captured a total of 2474 different vessels covered in multiple images, resulting in a total of 136,888 vessel bounding-box images. Multiple CNN detector architectures are evaluated in-depth. The SSD512 detector performs best with respect to its speed (85.0% Recall@95Precision at 20.1 frames per second). For the re-ID of vessels, a large portion of the total trajectory can be covered by the successful detections of the SSD model. The re-ID experiments start with a baseline single-image evaluation obtaining a score of 55.9% Rank-1 (49.7% mAP) for the existing TriNet network, while the available MGN model obtains 68.9% Rank-1 (62.6% mAP). The performance significantly increases with 5.6% Rank-1 (5.7% mAP) for MGN by applying matching with multiple images from a single vessel. When emphasizing more fine details by selecting only the largest bounding-box images, another 2.0% Rank-1 (1.4% mAP) is added. Application-specific optimizations such as travel-time selection and applying a cross-camera matching constraint further enhance the results, leading to a final 88.9% Rank-1 and 83.5% mAP performance.

## 1. Introduction

Water authorities are interested in vessel-speed enforcement systems, because of the necessity to bound Maritime traffic to speed regulations. The main motivation is safety, especially in areas where tourist and commercial vessels share the waterways. Higher waves generated by speeding vessels can cause safety risks for other waterway users and induce dangerous currents for swimmers or small vessels. In addition, these waves create an increase in water displacement, which results in increased erosion of the shoreline and structures in the water such as bridges and jetties.

Although commercial vessels have an incentive to take on higher speeds, because of cargo delivery times, they have measurement sensors aboard and they broadcast their ship ID, speed and location in a regular interval over the Automatic Identification System (AIS) radio system. This information is directly used for law enforcement because it provides accurate measurements in all weather conditions over a large range. Recreational vessels do not have such a system, rendering them invisible to law enforcement, and should be speed-monitored in a different way. The availability of these data thus depends on the willingness of commercial vessels to transmit such data to harbor authorities. Therefore, for law enforcement, it is attractive to install a visual sensor system for speed measurement of vessels in general.

Multiple systems exist for vessel detection in optical satellite images (survey by Kanjir et al. [1]), however, live satellite images are not always available. Camera surveillance systems are being deployed for vessels, but their installation is limited to sluices and important junctions of waterways or cargo loading areas, which are mostly individual systems with local installation without any speed measurement. To this end, we investigate a novel camera-based system for the application of vessel speed enforcement on waterways. Local speed measurement in a single camera view is difficult and requires accurate camera calibration and accurate position estimation of the moving objects, and is sensitive to small camera movements from wind. Therefore, speed measurement is typically performed over a longer trajectory.

In automobile traffic applications, the average speed of vehicles over a longer trajectory is typically implemented using re-identification between two sensor locations using embedded licence plate recognition. In contrast to vehicles, vessels do not have well-defined licence plates or other common visual registration markers. However, most vessels are unique based on their overall appearance, since there are many different vessel types, and the arrangement of bows, cabins or various other details such as flags or buoys, makes their appearance more unique. Therefore, vessel images captured at different camera locations, which enable to make a unique signature of the vessel appearance including the previously mentioned details, potentially allow re-identification of unique vessels. The previous description motivates the use of a water-side camera system which is suited for vessel re-identification.

This solution poses several challenges such as the fluctuating weather conditions, highly dynamic lighting conditions due to the constantly moving water surface, and occlusions between vessels. Another challenge is the large variation in vessel sizes, since vessels vary from small rowing boats to large barges of more than 100 m in length, covering the entire camera image.

In this paper, we propose a camera-based system that measures vessel speeds by detecting and linking vessels with two water-side cameras, placed several kilometers apart. The system detects and tracks vessels within each individual camera view. Then, vessels are linked from one camera to the most similar historic vessel images in the other camera by a re-identification system (further referred to in this paper as re-ID system). When a vessel is recognized and linked in both cameras, its speed is determined using the travel time compared to the distance between the two cameras. The distance between the cameras is known and the two systems are time-synchronized. Therefore, if a vessel is correctly recognized and linked in both cameras, its computed speed computation is accurate, and an additional validation of the estimated speed values is not performed. We specifically investigate three aspects of the vessel re-identification problem: (1) vessel detection and tracking in a single camera, (2) re-ID of vessels based on their visual appearance between two cameras, and (3) the creation of an image dataset required for training the recognition algorithms. The challenges for developing a camera-based re-ID system are multifold. First, the re-ID concepts for regular traffic participants have been evaluated successfully, but such re-ID has never been confirmed for vessel detection systems. Second, as already indicated, it is not sure whether the visual appearance properties forms a sufficiently unique pattern to perform re-ID with a high reliability. Third, the lighting and weather conditions are highly variable on water surfaces and there is a large variation in waterway scenery. These large variations and fluctuations may very well hamper successful usage of re-ID concepts. Some example images of a vessel entering and leaving a canal are shown in Figure 1. Note that both cameras are observing vessels from different directions, which complicates the re-ID problem but poses a realistic constraint for wide-scale deployment where physical constraints in the scene limit the choice of camera viewing direction.

For the investigation of such a vessel re-ID system, several developments of re-ID literature based on traffic of vehicles and persons have been presented with recent advances in deep learning, which make it likely that the proposed techniques can be also applied to vessel re-ID. The success of deep learning has made object detection tasks so robust that it is plausible to assume that they will also facilitate vessel re-ID despite the large variations. With respect to the recognition of vessels, the multi-view capturing of vessels seems an important feature to ensure that a unique appearance representation can be constructed. This feature will therefore be incorporated in the design of the new system. Dataset construction for vessels has not been a major point of interest and as far as data availability is concerned, the existing data are mainly concentrating on vessel detection and classification, rather than other vision tasks.

The research work on the design of a vessel re-ID system has resulted in the following main contributions.

A novel system is presented for speed measurement of vessels that combines detection, tracking and re-ID algorithms.A new dataset is created with a broad variation of vessel types and sizes to thoroughly validate the system concept. The dataset consists of 18,388 images with 1237 vessel trajectories that are linked between both cameras.Different CNN-based object detectors are compared and the training dataset on the vessel detection accuracy is investigated.An innovative combination of re-ID and tracklet concepts are jointly used for high re-ID accuracy. This total concept is well evaluated.A smart rule-set has been designed to improve overall re-ID performance and accuracy. These rules are based on application constraints such as the distance between the cameras (a-priory known) and the data characterization sharing of vessels between the two cameras. These rules significantly improve recognition performance and thereby the reliability of the system.

Compared to our previous work [2] presented at the VISAPP Conference (VISAPP Conference 2020, Valetta, Malta, http://www.visapp.visigrapp.org/ (accessed on 5 July 2021)), we have extended our experiments for detection by comparing with two other datasets and comparing different detection models. Furthermore, the re-ID experiments have been extended by including an additional re-ID model, several new extensions to our trajectory-based re-identification and an execution-time analysis.

The remainder of the paper is divided as follows. Section 2 introduces related work for all components of our system. Next, Section 3 describes the proposed system. The dataset used for the experiment is presented in Section 4. The experimental validation of the proposed system is divided in the evaluation of the vessel detection performance in Section 5 and the re-identification performance in Section 6. The paper finalizes with conclusions in Section 8.

## 2. Related Work

The proposed application consists of a complete pipeline for visual detection and re-identification of vessels. Limited work is available for vessel-speed enforcement with surveillance cameras.

### 2.1. Application-Oriented Work

One particular system is ARGOS [3], a vessel-traffic monitoring system in the city of Venice. This system employs surveillance cameras with slightly overlapping views, mounted high above the waterway. The authors employ background modeling for detection and tracking of all vessels.

The work of Qiauo et al. [4] concentrates on unrestricted detection, tracking and re-ID of moving vessels, to recognize other vessels in their surroundings. They use a CNN-based detection model [5] and apply sensor fusion and a re-ID algorithm to achieve accurate vessel tracking in their surroundings. The authors show that detection, tracking and re-ID is feasible for vessels, although they use re-ID only within a single camera view to improve the tracking algorithm. Ghahremani et al. [6] also successfully employ vessel re-ID within a single camera view.

The main components of our envisioned speed enforcement system are vessel detection and re-identification of vessels in non-overlapping camera views. Therefore, we discuss also related work on detection and re-identification in the following subsections.

### 2.2. Vessel Detection

In the generic field of visual object detection, state-of-the-art performance is achieved by Convolutional Neural Networks (CNNs). There are two main groups of CNN detection techniques. The first group splits the problem into a region proposal and a refinement stage. The region-proposal stage proposes bounding boxes that possibly contain an object of interest. These proposals are refined and classified into categories by the second stage [7]. These stages are combined into a single CNN [8,9] and can even perform instance segmentation in the refinement step [10]. The other common group of CNN detection techniques skips the region proposal step and estimates bounding boxes directly from the input image. Examples of such detectors are YOLO [11,12], Single Shot Multibox Detector (SSD) [13] and Fully Convolutional One-Stage Object Detection (FCOS) [14]. All detection systems use a baseline CNN architecture with an additional detector head to convert features into object bounding boxes. YOLO uses the top feature map to directly predict bounding boxes for each cell in a fixed grid. SSD extends this concept of using anchor boxes at the topmost feature map by adding anchor boxes with multiple aspect ratios and detector heads at several scaled versions of the top feature map. These anchor boxes serve as hyperparameters which are refined by estimating offsets compared to these boxes. FCOS adds a feature pyramid and proposes to avoid the use of anchor boxes altogether. For each feature layer element, it estimates a 4D vector that encodes the pixel-location of a bounding box at that feature layer. This results in more simple loss functions and less post-processing during inference. The computational complexity is higher than the other detectors due to the additional feature pyramid. Early work in detection of vessels uses handcrafted features and motion analysis [15,16]. However, at present CNN networks have proven consistently to outperform handcrafted features. Prior work by Zwemer et al. [17] shows that the SSD detector is robust against large-scale variations of vessels. We select the SSD detector for our application because of the relatively low computational requirements and high accuracy that has been proven for the vessel detection problem. However, for comparison purposes, we also evaluate the performance of YOLO, FCOS and Faster-RCNN implementations for our application.

### 2.3. Vessel Re-Identification

Related work for re-identification is mainly focused on the domain of person re-identification and we have not found any literature that specifically targets vessel re-identification. However, for the re-identification of road vehicles, recent work evaluates the performance of state-of-the-art re-ID algorithms [18], showing that these re-ID algorithms generalize to other domains. Re-ID systems for persons typically use a CNN architecture based on a ResNet-50 backbone, which is pre-trained on ImageNet. Considering that ImageNet has many classes, including persons, cars, trains, etc., it is expected that these re-ID algorithms do generalize well to other domains.

In the last decade, visual re-ID performance has been significantly improved by including aspects of recent developments in CNNs [19] and can be divided in two main techniques: pairwise verification and metric embedding. Pairwise verification networks are typically Siamese networks [20] that are trained on image pairs and try to estimate if the two images correspond to the same object. These networks learn to increase the feature distance between two different objects from their images, while decreasing the distance for two images of the same object. Alternatively, metric embedding networks consider so-called (image) triplets that are generated for each considered image in the training set (the anchor image). A triplet is constructed from the anchor image, an image from the same object and an image from a different object. Note that the corresponding triplet loss always simultaneously considers a matching and non-matching image pair, while the contrastive loss of pairwise verification networks considers just one image pair [21]. As a result, metric embedding networks utilize inter-class and intra-class variances in the resulting embedding space more efficiently. In this case, both classes with large and small intra-class variance will be nicely separated in the embedding space.

Another advantage of using (image) triplets is its effective hard triplet mining, called batch hard. This technique has been introduced by [21]. One of the main issues with training re-ID networks is a strong class-imbalance between matching samples and non-matching samples, namely all other vessels than the considered vessel are considered non-matches. Because most samples are quite different, it is important to select the most informative samples during training. With this batch-hard strategy, each image in a batch becomes the triplet anchor once, and the triplets are created by selecting the hardest matching and non-matching samples in that batch (hardest in the embedding space). Since the batch-hard strategy selects the hardest triplets possible in a random subset of the total training set, it provides a good balance between hard, nominal and trivial samples, which has proven beneficial for training [21]. For these reasons, metric-embedding networks are generally preferred and have been adopted in recent re-ID literature.

Another commonly used network-based technique is to focus on different parts of an object, such as the head, body and legs for persons. The current state-of-the-art model that integrates object parts is the Multiple Granularity Network (MGN) [22]. The CNN architecture of MGN constructs three separate branches on a backbone network, each focusing on a different partitioning of the object. The first branch considers the global person image, the second partitions the image into two parts (upper and lower body) and the third branch applies three partitions (head, torso, legs). In addition to the partitioning, the MGN network is trained using both the triplet loss and cross-entropy loss. The cross-entropy loss is typically used in training of classification networks and is added to the triplet loss to obtain a better optimization. Intuitively, this assists in distinguishing two different persons that are visually similar. Joint usage of these two loss functions has gained popularity in recent re-ID work [18,23,24].

Finally, there is also interesting re-ID literature on improving the effectiveness of the feature embeddings, by changing the matching procedure while using the same embedding methods. For example, Zhong et al.propose to use re-ranking, which cross-matches the set of most likely matches. More specifically, for a close match it verifies whether that image has the query image also in its set of most likely matches. If not, it is removed from the set of most likely matches of the current query. In previous work [25], we also refined the set of most likely matches for a query based on scene understanding, such as possible paths between cameras and plausible transition times. Consequently, this results into a significant gain in performance.

In our work, we focus on surveillance cameras with more dynamic backgrounds and our system utilizes cameras with non-overlapping views and located several kilometers apart. Another restriction is the required camera height to obtain a suitable birds-eye view of the scene. In open fields, cameras would require to be mounted on high poles, resulting in resonance and movement in the camera images, such that the output videos are not suitable for conventional computer vision techniques. In this paper, we propose to employ CNN-based analysis and given the related work on image triplets, we are interested in both the computationally efficient TriNet model [21] and the more complex and higher performing MGN model [22].

## 3. System Overview

The proposed system is divided in two stages (see Figure 2): camera processing, including vessel detection in each individual camera, and vessel re-identification that links vessels between the two cameras. The first stage performs real-time detection and tracking of vessels. For each recognized vessel, a set of vessel image crops and a timestamp is stored in the database and sent to the re-identification stage. These crops enable the construction of a unique feature vector description of the vessel that embeds the visual properties. The selection of the image crops of the same vessel positioned over a short period of time is the input for the re-ID stage. The final selection of image crops of the same vessel is called a tracklet of image crops (TIC). Note that the terms trajectory and tracklet refer to the same concept in this paper. The re-identification stage then checks if this vessel has also been recognized by the other camera, by evaluating all images stored in the gallery dataset of the other camera. All possible visually matching vessels are then evaluated using the travel time constraints that are imposed by the fixed distance between the cameras and the speed limit at hand. If a match is found for a vessel that traveled faster than the speed limit, this match is sent to the system operator that has to manually verify that the vessel is indeed identical. In the following subsections, the implementations and aspects of both stages and their embedded processing steps are discussed in more detail.

### 3.1. Camera Processing Stage

A. Vessel Detection:vessel detection is performed using the Single Shot Multibox Detector (SSD) [13]. This detector is based on a Convolutional Neural Network (CNN) and consists of a base network and a detection head. The base network converts the input image of size of 512×512 pixels into a feature representation. The detection head contains several convolutional layers which create downscaled versions of these features. The detection head predicts an object confidence and bounding box on each of these feature layers. The box is predicted using offsets to a set of fixed anchor boxes. The output of the detection head is created by combining the class confidences, location offsets and anchor boxes of all feature layers. These combinations are then filtered by thresholding the class confidences followed by non-maximum suppression, based on the Jaccard overlap (IOU) of the boxes, leading to the final set of detections.

To match the input resolution of the SSD detector network, a fixed square region in the camera image is selected as input for the detection network and scaled to 512×512 pixels. The video from the camera is processed by the detector at a rate of five frames per second. This results in bounding-box localizations of all detected vessel in each image.

B. Tracking: detected vessels are tracked over time at the rate of 25 frames per second. Every detected vessel is compared to existing tracks using a box overlap. If sufficient overlap occurs, the tracker is updated with the detected bounding-box coordinates. If no track exists for a detection, a new tracker is started. The detector can be applied at a somewhat lower frame rate (5 fps), so the tracking algorithm only needs to track objects over the intermediate frames (4 frames). Therefore, a simple and computationally efficient tracking algorithm can be used, so that the tracking algorithm does have a negligible performance impact on the complete system discussed in this paper. Tracking is performed independently for each object using feature point tracking [26] with optical flow. For each tracked vessel box, a uniform grid of feature points is created and new positions of each point are estimated in the next frame based on their estimated displacement by optical flow. The median displacement of the feature points determines the new position estimate of the vessel box. Over a period of 1 s, 5 detections are found with each of them 4 tracked positions in between. Tracking continues until the vessel leaves the scene.

C. Tracklet of Images Crops (TIC): the output of the tracking contains the group of boxes at tracked positions of the same vessel, starting with the first detection of that vessel until the vessel left. This group of boxes is then downsampled to 1 bounding box per second, beginning at the first detection. The downsampling inherently limits the amount of transmitted data to the re-ID stage. The result after detection and tracking for each vessel is a set of cropped vessel images over time, representing the tracklet of image crops (TIC).

### 3.2. Vessel Re-Identification (Re-ID) Stage

The re-ID stage receives vessel images of the recognized vessels from one camera and finds the corresponding vessel in historic images of all detected vessels (TIC Database). As shown in Figure 2, the re-ID stage consists of an image-embedding system (D), a TIC Database (E), a filtering system (F, G and H together) and a matching system (I). Our main novelty lies in the latter two systems, which are specifically tailored to our application. We have adopted two existing models for the image-embedding system: TriNet [21] and MGN [22]. In this subsection, we start by summarizing the used CNN architectures and discuss the common way of evaluation and implied matching procedure in re-ID literature. Afterwards, we elaborate on the proposed improvements concerning that matching procedure.

D. Feature Embedding: for the system responsible for creating the image embeddings, the existing networks TriNet [21] and MGN [22] are used. Independent of the applied model, they convert each vessel image into a feature vector that embeds the visual vessel properties into a relatively low-dimensional vector representation. First, we discuss the two adopted triplet-based models.

In the TriNet model [21], each vessel image is first scaled to a resolution of 288×144 pixels and then randomly cropped to 256×128 pixels to match the input of the ResNet-50 base network of the TriNet model. The last layer of the ResNet-50 model is replaced by two fully connected layers of 1024 and 128 units, respectively. By applying the TriNet model, every single image in a vessel trajectory is thus converted to this 128-dimensional embedding, which is subsequently stored in the database. The model is trained end-to-end and uses the triplet loss function to perform metric learning.

The MGN model [22] has a more extensive network architecture. First, it rescales the images to a resolution of 384×128 pixels prior to insertion into a partial ResNet-50 backbone. The backbone connects to three different branches and each branch consists of the remaining residual blocks of the ResNet-50 backbone (without shared weights). These branches all guide their training with differently connected loss terms, but all branches are trained using a triplet-loss and softmax-loss term on their global feature map. For the second and third branch, an additional two-way and three-way partitioning is applied based on the height dimension of this global feature map, respectively. The partitioning is performed such that these branches can also guide their training by connecting a softmax-loss term to each partition individually. Finally, by applying MGN at test time on a sample image, a 2048-dimensional embedding is created by concatenation of the embedding subparts that each branch produces.

The computational cost of the re-ID system is twofold. When a new vessel is recognized in a camera, a description of each vessel image is generated, resulting in a set of feature vector embeddings. Second, the vessel is compared to the database of vessels, previously observed in the other camera. For the comparison, a distance-measure is calculated between each new vessel embedding vector and the set of existing vectors in the database, where the amount of comparisons is equal to the number of vessel embedding vectors. The distance between cameras relates to the amount of vessels that are stored in the database. Depending on this configuration and the used re-ID model, the computational complexity is dominated by the calculation of the feature embedding or the computation of the similarity to the dataset entries. Note that the TriNet model creates 128-dimensional vectors, while MGN generates 2048 dimensions.

E. TIC Database: Once a model is trained, it learns to produce meaningful feature embeddings and its final re-ID performance is determined on the test set. For this, both a database of images with known vessel IDs, a so-called gallery set, and a query set is required. Similar to the most popular (person) re-ID datasets, DukeMTMC-reID [27] and Market-1501 [28], we define a (single) query as a newly observed vessel with unknown identity, where the query set is one random image for each vessel tracklet in the test set. Each vessel in the test set will be used once as a query, represented by a single random image from its tracklet. The gallery set is defined as the full test set with exclusion of the query set.

In contrast to using a single random sample, we represent a query by its whole tracklet. Consequently, we group the embeddings created by the Feature-embedding system (D) per TIC and store them in the TIC Database. These TIC-based feature embeddings are then used in our customized (tracklet-based) matching and filtering systems. For making comparisons, the common matching procedure uses a gallery without additional filtering, so that such a system would only have components D, E and I, where I is image-based. We propose to customize this common matching procedure by applying a total of four extensions that results in significant improvement of the matching performance. These four extensions are now defined, starting with the most important one for ease of explanation.

I. Tracklet-based matching. In this extension, we propose to perform matching by Tracklet-based Querying, where all images per query-vessel tracklet are utilized. In line with re-ID literature, we use the term tracklet to indicate an object trajectory in this work. To utilize the full tracklet, we accumulate the database similarity scores for all of its images to improve matching accuracy. This is substantially different from the common matching procedure, where just a single image is used per query-vessel tracklet to compute the similarity with the database images. As depicted in Figure 3, we match each image in the query tracklet individually with each image in the gallery set and, per gallery image, combine the resulting similarity scores over all different images in the query tracklet (by summation). The database image with the highest aggregated similarity score defines the most likely match.

F. Query tracklet filtering: Based on the image size of the images in the query tracklet, we remove a certain percentage of the tracklet by filtering, in order to reach the full potential of the Tracklet-based matching. This is beneficial for re-ID performance because the largest vessel images contain most fine details of the vessels and there are often many vessel images with too few details for accurate matching.

G. Vessel travel-time filtering in database: Based on the minimum and maximum possible travel time of vessels between the two cameras, we construct a different subset of the whole gallery for each query. For each query vessel, the gallery only contains those vessels that satisfy the timing constraints for passage from one camera to another. This increases re-ID performance because mismatching vessels are removed from the search set (the gallery subset).

H. Cross-camera filtering in database: Finally, in our application, vessels from Camera 1 only need to be matched with vessels from Camera 2, and vice versa. Hence, we ensure that the gallery only contains images from the other camera. It is common practice in re-ID literature that the gallery contains images from all cameras. Hence, to safeguard that our reported re-ID performance is well comparable with re-ID literature, we introduce this extension at the end of the experiments.

## 4. Vessel Datasets

### 4.1. Datasets for Detection

Popular datasets for generic visual object detection are MS-COCO [29], ImageNet [30] and PASCAL-VOC [31]. Although these datasets do contain images of vessels, they are taken from very different camera viewpoints not matching with typical surveillance scenarios. The dataset proposed in [17] consists of vessels in surveillance scenarios with multiple camera viewpoints and can be used for training a vessel detector. However, this dataset lacks vessel trajectories and identifications of the vessels, so it cannot be used for training the re-ID network. A large dataset of vessels called MARVEL [32] containing two million vessel images, focuses on classification and is based on only cropped vessel images without context, so that it cannot be used for training a vessel detector because all vessels are located at image center without bounding boxes. A similar dataset focusing on the same harbour application and containing images from various maritime areas named VCA-VCO has recently been made publicly available by Ghahremani et al. [33] (VCA-VCO dataset [33]: contact the authors to obtain a copy.). Figure 4a shows several example images from this dataset. Another publicly available dataset is the SeaShips dataset presented by Shao [34]. Although the paper claims that their dataset contains 31,455 images, the provided dataset for download contains 7000 images only (SeaShips-7000 dataset [34]: http://www.lmars.whu.edu.cn/prof_web/shaozhenfeng/E-publication.html (accessed on 5 July 2021). Therefore, we refer to this dataset as SeaShips-7000 in the remainder of this paper. Some example images of both are shown in Figure 4. Table 1 shows statistics on these datasets such as image resolution and the amount of images/objects available.

### 4.2. Datasets for Re-ID

Although both the VCA-VCO and SeaShips-7000 datasets consist of data for training a vessel detector, they do not contain any identification information and therefore cannot be used for training a re-ID system. The most commonly used datasets for re-identification focus on persons, such as DukeMTMC-reID [27] and Market-1501 [28], which can also not be applied for vessel re-ID. These datasets are closed sets, meaning that for each query object a representative object is available in the gallery set. Hence, to our knowledge, a suitable dataset for training a vessel re-ID system is not available. Consequently, we introduce a novel vessel dataset named ‘Vessel-reID’ containing trajectories and identifications. This dataset is used in this paper to train our vessel detector and re-ID system.

### 4.3. Vessel-reID Dataset

We introduce the Vessel-reID dataset containing vessels from two cameras, located 6 kilometers apart along a canal in the Netherlands. The set has been recorded over 4 days and contains 2474 vessels traveling between the two cameras. Each vessel is sampled over its full trajectory in the individual camera views by multiple images and the trajectories are linked between the two cameras. Figure 5 shows a schematic birds-eye view of the positions of the cameras. Note that the connecting canal is not covered by any camera (at the top in the figure). This causes some vessels to appear in one camera, but not reappear in the other camera. Fortunately, the majority of the vessels use the main canal and pass both cameras.

The Vessel-reID dataset has been created semi-automatically. For each video capturing day, we first process each camera stream individually to extract per-camera trajectories of vessels. The video is then processed with an existing vessel detector [17] to find vessels and track them over time through the camera view. The obtained vessel trajectories are temporally sub-sampled (every 6 s) and manually validated to enforce accurate localization and full coverage of all vessels. To produce the final dataset, interpolation of trajectories is performed to obtain annotations every second. To increase the annotation accuracy, the vessel detector is retrained with the annotations of the first day of video for both cameras, prior to applying it to the data of the other days.

After creating trajectories of the two individual cameras, the vessels in the two cameras are manually linked to define the re-ID ground truth. A few examples of different types of vessels and ground-truth annotations are shown in Figure 6. Figure 7 shows examples of linked trajectories of vessels in Cameras 1 and 2. Of all linked trajectories, the slowest vessel moved at 2.25 km/h, while the fastest vessel travelled at 23 km/h. An overview of all travel times is given in Figure 8b. The speed limit in the canal is 12.5 km/h. Since only a few vessels travel faster than the speed limit, we used all vessels for our experiments in determining speeding (max. 160 min travel time). For re-ID purposes, we have created a closed set and only use linked vessels as queries, meaning that a matching vessel is always available in the gallery for every query vessel.

In total, we annotated 4 days of video from 6.00 AM until 9.00 PM from both cameras. This resulted in a total of 136,888 vessel images containing 2474 trajectories (TICs) of 1237 unique vessels moving through both cameras. Each vessel trajectory was quantized at the rate of one sample per second. On average, there were 44 samples for each vessel TIC in Camera 1 and 66 samples in Camera 2. The different numbers can be explained by the different viewing angles of both cameras, resulting in different track lengths (see Figure 1). Table 2 gives an overview of the vessel trajectories per day. Note that about 30% of the unique vessels were moving into the side canal, mooring at a local harbour or appearing in one camera only. Since vessels that passed both cameras led to two TICs, the overall total of unique vessels in our dataset was less than the overall total TICs. Note that the two per-camera total TIC counts in Table 2 thus each incorporated the first two rows of that table plus some other rows of that table corresponding with that camera. For re-identification, this means that a total of 3013 TICs occurred in 4 days (both cameras), which led to an average required re-identification throughput of 50.22 TICs per hour. During the most busy hour in these 4 days, a maximum of 184 TICs per hour occurred for both cameras.

The dataset was used for training and evaluation of our detector and re-identification stage. To this end, the dataset was split into a training set containing the data of the first 3 days and a testing set containing the fourth day of data (see Table 1 for the overall statistics). By doing so, the test set contained true unseen data.

## 5. Experimental Results: Detector

Four main experiments were conducted to evaluate the performance of vessel detection. First, the effect of the training dataset on the detection performance was investigated. Second, four different CNN detection models were evaluated. Third, a more in-depth measurement with respect to the vessel sizes was performed using the same detectors. The final experiment gave insight in the robustness of the detection performance for re-identification purposes, by measuring the percentage of each vessel trajectory that was detected.

Detection performance evaluation was carried out using the same metric in each of the experiments. Performance was reported by recall-precision curves and summarized in a single value by the the mean Average Precision (mAP) metric, which averaged interpolated precision values of the positive samples. Recall R(c) denotes the fraction of objects that were detected with a confidence score of at least *c*. An object was detected if its bounding box has a minimum Jaccard index (Intersection-over-Union ratio) of 0.5, when compared with its ground-truth bounding box. Lower overlap values resulted in incorrect detections. Precision was defined as the fraction of correct detections. Hence, each point in the recall-precision curve represented a certain fraction of correct detections (precision) vs. the fraction of detected vessels (recall) for a certain detector threshold *c*. For our application in vessel detection, a working point on the recall-precision curve had to be selected. Therefore, we also report the obtained recall at 95% precision, since a high precision was desired and preferred over a high recall.

In all our experiments on the Vessel-reID dataset, we used the data subsets Days 1, 2 and 3 for training and Day 4 for testing. All detectors were trained on the manually validated frames in the dataset only (every 6 s of an annotated vessel trajectory). The SSD detector was trained by fine-tuning the VGG16 base network from the original paper [13], using the default anchor-box configuration. We performed 10,000 warm-up iterations (linear) of ratio 0.0001 and used an SGD optimizer with a learning rate of 0.02, momentum of 0.9 and weight decay of 0.0005. Training was performed with a batch size of 8 for 24 epochs (during one epoch we sampled the dataset five times, resulting in ≈16,500 iterations per epoch). The learning rate was decreased at epochs 16 and 22.

### 5.1. Cross-Validation on Datasets

This experiment investigated the effect of the dataset on the detection accuracy by performing a cross-validation between the different datasets. The SSD detector was trained four times: on each dataset individually and on all datasets combined. Evaluation was carried out on each dataset separately. The datasets used in this experiment were SeaShips-7000, VCA-VCO and our Vessel-reID set. We reported average recall-precision curves and standard deviations by evaluating over three training cycles. The results are presented in Figure 9.

SeaShips-7000 (Figure 9a). Best performance was obtained when training with SeaShips or the combined dataset, resulting in very high 99% mAP scores. When training with VCA-VCO, the performance was significantly lower because of the larger variation of vessels: the precision dropped under 90% at around 46% recall. The detector trained with our Vessel-reID dataset had poor performance (17.1% mAP score). We concluded that all datasets were quite different from each other (visually mutually exclusive, which will be discussed later) and that when only one of the existing datasets was used for training, it lead to a significant reduction in detection accuracy.

VCA-VCO (Figure 9b). We observed a similar pattern as compared to testing on the SeaShips-7000 dataset. The VCA-VCO dataset apparently was quite different from the other training sets. The detectors trained on the VCA-VCO dataset and the combined sets both had high detection performance (96.8% and 96.6% mAP, respectively). The detector trained on SeaShips-7000 had a low performance (42.4% mAP), as precision dropped under 90% already at below 10% recall. The detector trained on our dataset performs even worse, obtaining only 29.2% mAP score.

Vessel-reID (Figure 9c). A similar behaviour was observed again, training with the Vessel-reID set and the combined set results in an mAP score of 89.8% and 90.2%, respectively. The detector trained only on VCA-VCO data has a reasonable performance (76.7% mAP) with about 65% recall at 90% precision. The precision drops faster for the detector trained on SeaShips-7000, but this detector still obtains an mAP score of 57.1%.

Overall, the results show that the three datasets were visually quite different. Training on the same dataset always results in best performance, while training with all datasets results in a comparable score. SeaShips and Vessel-reID seemed to be the most different datasets. The VCA-VCO dataset was the most diverse dataset, as a detector trained on this dataset always obtains second-best performance on the two other datasets. In all cases, training with all datasets did not hurt performance, suggesting that the different datasets did not visually overlap much. We concluded that it was best to train on the combination of all datasets to obtain the most generalizing detector that also performed well on each individual dataset. In the remainder of the experiments, we therefore used a detector that was trained on the combination of all datasets.

### 5.2. Detector Comparison between CNN Implementations

To provide insight in the performance of the SSD detector, we present a comparison between different CNN detection models. In this experiment, a comparison was made between SSD [13], YOLO-v3 [12], FCOS [14] and Faster R-CNN [9]. The YOLO-v3 model used the Darknet-53 backbone and was trained on input images with a resolution of 608×608 pixels. FCOS and Faster R-CNN both use the Resnet-50 backbone and worked on an image input resolution of 1333×800 pixels. FCOS was trained with Deformable ConvNets v2 in the backbone and head. Each detector was fine-tuned with their default training parameters (Training parameters: https://github.com/open-mmlab/mmdetection/tree/master/configs/ (accessed on 3 March 2021)) for 24 epochs. The detectors were all trained on the combined data from SeaShips-7000, VCA-VCO and Vessel-reID. The detector performance was measured by the mAP score and the recall-precision curve on the test sets of each dataset.

The obtained detection performance on the three datasets was depicted in Figure 10.

SeaShips-7000 (Figure 10a). The recall-precision curves on the SeaShips-7000 dataset show that Faster-RCNN and FCOS had the highest performance, reaching a high mAP score of 99.7% and 99.3%, respectively. The recall-precision curve of the SSD512 detector was slightly lower, achieving 98.5% mAP score. The YOLO-v3 detector has lower performance (92.7% mAP) and the precision dropped earlier.

VCA-VCO (Figure 10b). Similar observations could be made for the VCA-VCO dataset. Faster-RCNN, FCOS and SSD512 all seemed to have similar curves. Faster-RCNN has the highest overall precision. The precision of FCOS dropped slightly at an early stage (at 10% recall). However, it surpassed the precision of SSD512 at about 83% recall. The precision of the YOLO-v3 detector started to decrease at 10% recall. However, YOLO-v3 still obtained a respectable 89.4% mAP score.

Vessel-reID (Figure 10c). The SSD512 detector achieved the highest mAP score with respect to the other detectors (90.2%), followed by Faster-RCNN (85.8% mAP), YOLO-v3 (85.6% mAP) and FCOS (78.3% mAP). This was mainly caused by the SSD512 detector obtaining a higher recall, whereas the precision of Faster-RCNN was slightly higher. However, the final recall of Faster-RCNN was only 86%, effectively resulting in a lower mAP score. The recall-precision curve of FCOS ended at just under 80% recall, causing the low mAP score. The curve of YOLO-v3 showed lower overall precision, but achieved higher recall than Faster-RCNN and FCOS.

Overall, the performances of Faster-RCNN, FCOS and SSD512 were all in the same range, while YOLO-v3 obtains the lowest accuracy. Although the results on the SeaShips-7000 and VCA-VCO dataset were consistent, on the proposed Vessel-reID dataset the results were slightly different. We can explain this by the fact that the proposed dataset was automatically annotated by another SSD detector model (as explained in Section 4). The dataset was manually edited after automatic detection during the dataset generation process. As a consequence, false detection boxes were manually removed and incorrectly localized annotations and missed annotations were edited. Despite this manual correction, the annotations in the dataset may have left a small bias to favor the detection results of the SSD detector.

### 5.3. Detection Performance with Respect to Vessel Size

In this experiment, a more in-depth inspection was carried out based on the vessel size. As vessels varied in size, from small inflatable boats to large barges, it was interesting to measure whether vessels of all sizes were detected. Therefore, the detection performance with respect to the ground-truth sizes was measured on each dataset. The ground-truth bounding boxes of each dataset were split into the following bins based on their areas: extra small (XS, a bounding-box area belongs to the smallest 10% of all bounding-box areas in a dataset), small (S, between 10$ and 30%), medium (M, between 30% and 70%), large (L, between 70% and 90%) and extra large (XL, 90%+). The actual area sizes of each bin are presented in Table 3. Each detection was matched to a corresponding ground-truth bounding box, and thereby categorized into one of the bins. If a detection was a false positive and could not be matched to a ground-truth box, it was categorized into a size bin according to its area. This area evaluation was similar to the COCO [29] implementation, but we extended the amount of size bins to capture the large variation in vessel sizes. The evaluation was performed separately on the three datasets, for each of the detectors trained in the previous experiment. The results are depicted in Figure 11 and some visual detection examples are illustrated in Figure 12.

SeaShips-7000 (Figure 11a). All four detectors had a quite stable detection performance over all vessel sizes. Only SSD512 and YOLO-v3 showed a small drop in performance for extra small vessels. It can be observed from Table 3 that the sizes of the vessels in SeaShips-7000 were relatively large, and the spread in sizes was not as large as in the VCA-VCO and Vessel-reID datasets.

VCA-VCO (Figure 11b). It can be observed that for all detectors, performance dropped for extra small ships. The lowest performance was obtained by YOLO-v3 at 58% mAP. The SSD512 detector results in a large drop from 95% mAP for small vessels (S category) to 82% mAP for XS vessels. The FCOS and Faster-RCNN detectors both had a small drop in performance for XS vessels of about 0.5% mAP, and were more robust at detecting the smallest ships than SSD512 and YOLO-v3.

Vessel-reID (Figure 11c). The drop in performance for small ships was even more significant on the Vessel-reID dataset. This can be explained by the extreme variations in vessel sizes with respect to the other datasets (see Table 3). Faster-RCNN offered the highest performance on XS vessels (62% mAP) followed by YOLO-v3 (55% mAP), FCOS (50% mAP) and SSD512 (44% mAP). Although SSD512 performed worst on XS vessels, there was a significant performance increase for small (S) vessels, moreover together with Faster-RCNN it performed best on small vessels (91% mAP). FCOS gave lowest performance for small ships (84% mAP), followed by YOLO-v3 (89% mAP).

Overall, these results show that all detector models obtained good performance for small-to-medium-sized vessels (S-M), but struggled with very small vessels (XS, S). The Faster-RCNN detection model seemed most robust to very small vessels. Large vessels did not pose a problem for any of the detectors, since in general the detection performance increased for larger vessels. Investigating some visual examples (Figure 12), it can be observed that Faster-RCNN provided a very accurate detection. Interestingly, Faster-RCNN could detect sailing ships with high masts quite accurately, while the other detectors had difficulties in finding such vessels. In addition, all single-shot detectors had difficulties detecting vessels that were partly occluding each other. Note that the large barges (at the top row) should both have been detected as two separate vessels: the barge and the pushing vessel. We expect that adding more training samples for such specific cases will improve the detection accuracy in general for all individual models.

### 5.4. Detection Execution-Time Analysis

In this experiment, we measured the throughput of each of the CNN detection models. The execution was carried out on a single GeForce GTX 1080 GPU for a batch size of unity. Timing was averaged over 2000 forward inference cycles. The timing was measured in (detection) frames per second and was plotted with respect to the recall score of each detector at 95% precision.

Figure 13 shows the trade-off between execution time and accuracy. The YOLOv3 detector was clearly the fastest detector operating at 38 frames per second. It was followed by the SSD512 detector executing at 20.1 fps, Faster-RCNN at 13 fps and FCOS at 12.4 fps. For our application, we chose the SSD512 detector because it provided a high accuracy of 84.0% recall at 95% precision, while being able to sufficiently fast perform the inference processing.

### 5.5. Detection for Re-Identification

In our application of re-ID, it was important that each vessel was detected at least once in both cameras. Our testing dataset contained several images of each vessel along its trajectory, enabling the measurement of the amount of detections per vessel over its trajectory. In this experiment, we measured the detection ratio for each vessel which was computed by the amount of correct detections with respect to the amount of ground-truth annotations. This detection ratio was reported as a histogram over all vessel trajectories per camera location in our dataset. The experiment was performed using the SSD512 detector. We selected a sensitivity threshold at 95% precision, to limit the amount of false detections while still having a high recall of 84.2% (see SSD512 in Figure 10c).

Figure 14 shows the results per individual camera. Note that the vertical axis has a logarithmic scale. In general, many vessels were detected over more than 90% of their trajectory and almost all vessels had a trajectory coverage of more than 60%. Unfortunately, there were few vessels (11 in Camera 1 and 6 in Camera 2) that were detected in less than 10% of their available trajectory images. Visual inspection of these ‘missed’ vessels shows that these detections were typically very small boats (such as ‘dinghies’ behind larger boats) or boats moving close together (see Figure 12 ).

This experiment shows that the proposed detector obtained a high detection accuracy and delivers a dense set of image samples for the considered re-identification sub-system. Note that we did not investigate object tracking in this experiment and purely focus on detection. In the actual application, an early detection of a vessel at the beginning of its trajectory will therefore effectively lead to a larger measured part of its trajectory. For re-identification, it was important to detect each vessel at least once. Although a few vessels were encountered that were never detected, they were mostly attached to a larger boat (‘dinghies’), which did not influence the re-identification performance, since in those cases the main vessel was detected (See Figure 12f). Missed vessels that were close to each other require the combined vessel pair to be detected for correct re-identification in both cameras.

## 6. Experimental Results: Re-Identification

The re-identification stage was also experimentally validated. This section first presents the evaluation criteria (Section 6.1) and the related common matching procedure from literature (Section 6.2). Then, the results on our dataset are presented using a baseline re-ID system based on TriNet and MGN (Section 6.3).

In this research, instead of the common matching procedure, a new more-effective matching approach was introduced. The common procedure was practically adopted in all re-ID literature, because it was predefined by the most popular re-ID datasets as a part of their evaluation procedure. The common procedure was enhanced with various incremental improvements, while directly adapting the feature-embedding system to our application. Experimental results are presented for various querying approaches using tracklets, and results for additional filtering based on application constraints. First, Section 6.4 introduces the proposed Tracklet-based Querying Approach, followed by Section 6.5 describing the execution-time analysis and related improvements obtained by linear subsampling. Next, the advanced query tracklet filtering technique for the proposed Tracklet-based Query Approach (Section 6.6) was evaluated, followed by the gallery filtering based on implied transition times (Section 6.7), and finally the proposed ’only cross-camera’ gallery approach (Section 6.8).

### 6.1. Evaluation Procedure and Performance Metrics

The re-ID performance was reported using the two prevalently-used metrics: Rank-1 and mean average precision (mAP). The performance values are expressed as mean and standard deviation over 10 training cycles. Note that in all experiments, the same trained re-ID models were used in the evaluation and test with every vessel trajectory as a query to obtain comparable scores. In order to determine these performance metrics, the performed matching procedure is explained in Section 3.2.

Evaluation of a single query image results in a similarity score for each gallery image. Next, the gallery was ranked, meaning that it was ordered to similarity score in decreasing order with the most similar image being on top. The Rank-1 score then indicated the fraction of all queries that had a true match at the top (Position 1) of their ranking. Similarly, the Rank-10 was the fraction of all queries that had at least one true match at Positions 1 through 10 of their ranking. The mAP was calculated as follows. First, the average precision value of each individual query was calculated, which was measured as the average precision@k for only those images in the ranking that were actual matches, where *k* was the position of the match in the ranking. The mAP was then defined as the mean of each query average precision. More details can be found in [28,35]. In contrast to the Rank score, the mAP includes the position of all correct matches in the ranking. When all true matches were at the top of the ranking, this yielded 100% mAP. If the top-x of the ranking consisted of true matches interchanged with mismatches, the mAP reported a lower value, while the Rank-x metric reports 100%. Important here was that a mismatch degraded the mAP more when it occurred at the top than when it was lower in the ranking. The Rank-1 metric was the most important metric for the application of vessel-speed enforcement, since the system should automatically detect a single correspondence between two cameras to observe a speed violation and thus only the top of the ranking mattered. Throughout this section, we report the Rank-1 and mAP evaluation scores.

### 6.2. Default Query Approach from Literature

The default matching procedure that is commonly applied in re-ID literature is now discussed, since this introduces important background information for the proposed re-ID matching system. After the model was trained on the train set, this common matching procedure was applied to determine the Rank-1 and mAP performance of the re-ID model on the test set. This matching procedure was used by the evaluation procedure of most re-ID datasets [27,28], which explains why it was incorporated here.

Prior to matching, each image in the test set was first converted into an image embedding vector by the re-ID model, where the test set was always separated into a query and gallery set. The query set contained one random image from the vessel trajectory of each vessel in the test set, while all other images of those trajectories were added to the gallery set. Matching in the Default Query Approach was now carried out for each individual query image in the query set, by computing the Euclidean distance between the single query embedding and each gallery embedding. This distance resulted in a similarity score for each gallery image, which was then used to compute the Rank-1 and mAP scores. Important here is that each query vessel was represented by only one image from its tracklet.

Finally, note that the gallery contained all images of all cameras, including the camera of the query. An advantage of this approach was that a larger dataset was obtained for evaluation (with more mismatching samples), resulting in a better statistical validation of the algorithmic performance. However, it must be ensured that images from the same tracklet as the currently considered query did not count as possible matches, since these images were very similar in appearance and thus otherwise would appear at the top of the ranking. This was easily solved by applying a so-called excluder, which temporarily excluded those images from the gallery during the individual evaluation of each query.

### 6.3. Image Embedding Networks and Their Baseline Performance on Our Vessel-reID Dataset

This subsection outlines the first experiment, where existing re-ID models were evaluated using the Default Query Approach. This experiment will leverage the baseline performance of those standard re-ID models. The existing TriNet [21] and MGN [22] re-ID models were trained on the training set of our Vessel-reID dataset. Besides as using these trained models as reference (baseline), they were also used as image embedding for the matching procedure in the proposed overall re-ID stage (see Section 3.2 (D)).

Then, the models were used to construct image embeddings for all images in the test set. In this experiment, we started with optimizing the influence of the hyper-parameters for training of the models for our vessel application. In line with re-ID literature, the performance was evaluated using the image-based Default Query Approach (see Section 6.2).

The optimal training parameters for TriNet were investigated in previous work [2], leading to a learning-rate of $3\times 10^{-4}$ and 500 training epochs, while exponentially decaying the learning rate with a base of $1\times 10^{-3}$ starting at epoch 300. The system trained with these parameters resulted in a Rank-1 performance of 55.9% and 49.7% mAP on the Vessel-reID dataset, when applying the Default Query Approach. Similarly, we also optimized MGN and found that training the model with a learning rate of $2\times 10^{-4}$ for 1000 epochs with a learning rate decay of a factor 10 at epoch 400 and 650, to be optimal. Training MGN with these hyper-parameter settings on the Vessel-reID dataset resulted in 68.9% Rank-1 and 62.6% mAP scores. The reported settings for all subsequent experiments in the remainder of this study were based on these optimized settings for TriNet and MGN (baseline performance).

In comparison with public person re-ID datasets, both re-ID models showed a large performance gap with our vessel dataset. That is, on DukeMTMC-reID, TriNet and MGN achieved 75.4% [36] and 88.7% [22] Rank-1, respectively. On Market-1501, the scores were even higher with 84.9% [21,36] and 95.7% [22] Rank-1, respectively. The lower performance with our dataset is mainly explained by the hard conditions of our train and test sets. In the Vessel-reID dataset, the test set was recorded during a completely different day with different lighting and weather conditions, while the person re-ID datasets applied a random division into train/test sets, such that both sets had samples of the complete recording time range. Consequently, this resulted in a more similar train and test set, whereas in our dataset, the test set contained truly new unseen data. Furthermore, we observed that our dataset contained several vessels of identical brand and model (identical rental boats), which were very similar in appearance, but yet had a different vessel ID in our dataset.

### 6.4. Tracklet-Based Querying Approach

The baseline image-based Default Query Approach was now extended towards exploiting information from multiple images per vessel trajectory. This was expected to improve performance in cases where the randomly selected image from the query-vessel trajectory was a poor representative, e.g., a very low-resolution image (vessel far away from the camera) or a partial occlusion by another vessel occurred. To overcome this limitation, we considered a tracklet as a query instead of a single image in our Tracklet-based Querying Approach (see Section 3.2 (I)). The Tracklet-based Querying Approach applied inference on every image in a vessel tracklet and combined the per-image similarity scores. As a consequence, this also required changing the final evaluation (and matching) procedure, because the common query/gallery division approach dictated using the whole test set minus the query set as the gallery set. With this division approach, using each image of a tracklet instead of a single image would lead to an empty gallery set. Therefore, we proposed to keep all images in the gallery and consider one tracklet at a time as the query. To this end, we carefully validated that using the full test set as the gallery had a negligible impact on re-ID performance. Furthermore, adoption of the excluder, as explained in Section 6.2, ensured fair use of the gallery, so that the re-ID model could not select any query-tracklet images as a possible match in the gallery.

Figure 15 shows an example embedding distances matrix between several query tracklets and their three closest matching gallery tracklets (distance colors explained in caption). It can be observed that some query images were not a good representation for a query vessel, resulting in high distance scores (horizontal yellow bars). This was similar for some images in a gallery tracklet (vertical yellow bars). However, most query and gallery images of a matching tracklet pair result in low distance scores indicating that the use of tracklets for matching was beneficial for re-ID.

As mentioned in Section 3.2 (I) and shown in Figure 3, all possible similarity scores per gallery image were combined. The method used for combining was evaluated in the following two ways to find the best combining approach. First, we take the sum of *all* per-gallery-image distances (over each column). Second, we took the sum of *only the top-X* per-gallery-image distances (over each column), effectively only summing the three yellow entries in Figure 3, where X=3 is visualized as an example. In essence, the second approach allowed us to only consider the *Top-X* most representative images for the tracklet query. Intuitively, this enabled us to discard the less-representative images in the evaluation (low-quality or occlusion). Thereafter, the resulting aggregated similarity vector was used to compute the Rank-1 and mAP scores as introduced in Section 6.1. Note that the aggregated similarity vector was comparable to the vector from the Default Query Approach, enabling a direct comparison using the Rank-1 and mAP scores.

The effect of the number of selected matches (*Top-X*) was investigated in a fine-grained sweep over the range X=1 to 40, while its impact on re-ID performance is reported. For reference, all available matches (*All*) is also included. This optimization was performed on MGN, since it was our best-performing baseline algorithm. The results are presented in Figure 16. The performance increased gradually with increasing values of *X*, while it converged for *X* values of about 20–30. Values exceeding 30 resulted in a minor decrease in performance and eventually reach the same performance as the *All* method, when further increasing *X*. Note that up to the Top-15 setting, the performance was lower than when using all images (*All*). Clearly, there was a trade-off involved in focusing on too few versus too many most-representative images. From the optimal range X=20−30, where on Rank-1 X=20 achieved lowest spread among training cycles while X=30 achieved highest performance, we selected the Top-25 as our preferred setting. Furthermore, it performed well on both the Rank-1 and mAP evaluation scores. Moreover, larger values were not desired because selecting more images for a tracklet could potentially include less representative images (low-quality or occlusions).

Summarizing, for the best-performing MGN model, matching all tracklet images (*All*) results in a Rank-1 score of 73.9%. Using the *Top-X* method, a similar performance was obtained when selecting only the *Top-10* scoring images. The performance further increased when selecting the *Top-15* (74.3%) and *Top-25* (74.5%). The use of multi-image matching was clearly beneficial because it offers a total gain of 5.6% compared to the baseline performance (single-image matching, see Section 6.3).

### 6.5. Execution-Time Analysis and Related Improvements by Linear Subsampling

In this subsection, an analysis of the computational cost of our extended re-ID system is provided. The total computational costs consisted of two parts: computation of the embedding vectors and computation of the matching. Every time a new vessel entered the camera view and a trajectory was formed by the detection and tracking system, the unique vessel description for re-ID was generated by the calculation of the vessel embedding vectors. To perform re-ID, the system requires the embeddings for all images in the gallery, of all trajectories contained, and then starts the matching for this new vessel against the gallery. The embeddings for all images of each vessel were calculated once, and used both initially as a query and were then added to the gallery for future comparisons. Note that this approach avoided the calculation of all embeddings of the complete gallery at test time. Therefore, the computational cost of the creation of the embeddings was similar for both our Tracklet-based and the Default Query Approach.

After embedding, the matching was then performed by comparing the vessel similarity (embedding distance) with all previously observed vessel samples. Therefore, the computation cost for matching was linear to the number of vessel images in the gallery, which was linear to the number of vessels (assuming a constant number of images per vessel). Similarly, for embedding, the computation cost was thus linear to only the number of images in the vessel trajectory. Therefore, reducing the number of stored images per trajectory effectively reduced computation cost for both the embedding and matching phase.

In our Vessel-reID dataset, the average trajectory contained 63 images, increasing to over 300 images for slow-moving vessels, see also Figure 8a. Note that many of these images were visually similar, especially for slowly moving vessels, because of the minimal movement between samples. We therefore gradually reduced the number of images per trajectory using linear subsampling to a fixed amount and investigated its impact on the re-ID performance and processing time. As a result, the computation time for both embedding and matching of each vessel became now fixed, even for slowly moving vessels.

The results of applying linear subsampling are presented in Figure 17, where we indicate the re-ID performance and execution time of both the Default and our Tracklet-based Querying Approach. The execution times were measured using a naive GPU implementation on a system with a GeForce GTX 1080 GPU and a Xeon E5-2650 v4 CPU, where both the embedding of the query images and the distance calculations were performed on the GPU. The shown execution times are based on parallel execution on the GPU. As can be observed in the figure, there was virtually no decrease in re-ID performance when limiting the trajectories to roughly 20–30 images only. This reveals that on average, only about 20–30 images of a vessel trajectory were sufficiently informative for re-ID while the additional images were either similar or were less representative (low-quality or occluded). We proposed to select 20 images per trajectory as a good trade-off for optimal re-ID performance at a large reduction of the computation time, with a factor 3 (from 667 to 216 msec) and 5 (from 1250 to 275 msec) for the Default and the Tracklet-based Query Approach, respectively. For the best-performing Tracklet-based matching, the total computation time of 275 msec at 20 images/tracklet was divided in the embedding (212 msec) and distance computation (63 msec). With this configuration, we obtained a speedup of five times compared to the baseline re-ID system (1250 msec) without loss in Rank-1 accuracy and only a minor reduction of 0.3% in mAP. When comparing the proposed Tracklet-based Querying with the Default Querying at 20 images/tracklet, the accuracy increased with 5.3% Rank-1 for a relatively low increase in computational cost, going from 216 to 275 msec (+27%).

Furthermore, the proposed system obtained a throughput rate of 3.6 TICs per second for matching, which was more than sufficient for the use case considered. In our dataset, an average of only 50.22 vessels passed both camera views per hour (1 TIC every 72 s) with a peak of 184 vessels/hour (1 TIC every 20 s). In practice, even when our peak throughput occurred continuously and a vessel passed by every 20 s, the gallery could ultimately contain and allow us to match vessels that were observed at the latest preceding 45 days. In comparison, the total computation time of 275 msec was achieved when the gallery contained 7 h of data with vessels. Given the size of the overhead, it is therefore likely that the proposed system could even operate at real time on an embedded computing platform.

We also evaluated a more advanced subsampling approach that involved top-7 ranking results, but then refining that to top-3 adaptively on the basis of the incorporated embedding distance values. However, this approach did not result in any notable performance differences, even when changing the numbers 3 and 7 to other alternative numbers.

### 6.6. Advanced Query Tracklet Filtering for the Proposed Tracklet-Based Query Approach

For the Query-tracklet filtering system (see Section 3.2 (F)), we were inspired by the large spread in image sizes in the vessel trajectory samples and hypothesize that not all images in a tracklet were equally informative. Vessel images were smaller in pixel-count when vessels were further away from the camera and were larger in size when being closer to the camera. Evidently, this effect varied with the camera view configuration. We evaluated the assumption that vessel images with finer details were more informative for re-ID and thus that larger object visibility (bounding boxes) in the images improve re-ID performance. Moreover, when the vessel was entering the camera view, it was only partially visible and therefore resulted in an incomplete description for re-ID. We tested our hypothesis by excluding some of the data and involving a fraction of the remaining query tracklet data in two different ways prior to starting our Tracklet-based Query Approach. First, the middle part of the tracklet was emphasized by removing the beginning and end parts of the tracklet, thereby eliminating the samples where the vessel was potentially partially visible. In the second way, the largest bounding-box images in the tracklet were emphasized by removing the smallest samples (measured by pixel-count/bounding-box size), such that only the samples with most details remained.

The mean Average Query Image Size (mAQuIS) metric was included in the measurements to indicate how the bounding-box size changed when the remaining data fraction increased. This mAQuIS metric was computed in two steps, by first determining the average bounding-box image size of each query tracklet. Second, this was repeated for all query tracklets and then the mean over all averages per tracklet was computed as well. This mAQuIS metric was chosen to give a more meaningful result, since it focuses on a computation at a per query-tracklet level, similarly as the mAP was computed by its double mean. Consequently, since a small set of extreme outliers were most likely related to a single query-tracklet, these extreme outliers had less impact on the final mean value because each outlier now only influenced an individual tracklet average.

Figure 18 depicts the results on re-ID performance for both ways to reduce the remaining data fractions. Figure 18a shows the effect of focusing on the middle part of the trajectories. It can be observed in the figure that the re-ID performance was constantly decreasing when reducing the remaining data fraction from the tracklet. This implies that the beginning and ending parts of the tracklet were important. Figure 18b indicates how keeping the largest bounding-box images in a tracklet affected performance. In contrast to removing the tracklet end and beginning, reducing the number of bounding-box images increased performance here, because visibility of the vessels was augmenting. This confirms our assumption that more relevant information for re-ID was contained in the largest bounding-box images. Consequently, this way for defining remaining tracklet data was applied in our Query-tracklet filtering system (Figure 2f). The bounding-box size was more important than filtering images at trajectory beginning/end, especially since the largest bounding boxes always occur at the beginning or ending of a trajectory. This aspect also explains the decreasing performance when filtering the begin/end of tracklet data (Figure 18a). Note that the filtering of begin/end trajectory had a different effect on small/far away vessels and large barges. Large barges resulted in many images in which the vessel was only partially visible, while for small/far away vessels there were only few images with partial visibility.

Additionally, the mAQuIS curves supported the conclusion that bounding-box size was important. For example, when comparing both mAQuIS values at the 95% remaining data-fraction point, it has a value of 570 pixels for Figure 18a versus 579 pixels in Figure 18b. Hence, even though there was an increasing trend in the beginning of the reduction of the remaining fraction in Figure 18a, the mAQuIS curve was always lower than that of Figure 18b. Nonetheless, the increasing trend at the left of the mAQuIS curve in Figure 18a may seem special, but this can be readily explained by the viewing angle of the cameras. That is, even if a vessel traveled at constant speed, from the perspective of the camera it would move slowly when being far away and quickly when being close-by. Hence, its trajectory would contain mostly small samples and only a small fraction of the largest samples. As such, the average bounding-box size of both the very start and very end of the trajectory was lower than the overall average at first, causing the initial rise.

As a conclusion, we adopted the second approach for preserving tracklet data when some parts were omitted, where the focus will be on keeping the largest bounding-box images in our Query-tracklet filtering system. We chose a fraction of 40% as the preferred value for the fraction of remaining tracklet data, since this was a good balance between re-ID performance while keeping a sufficiently large number of bounding-box images to dominate the amount of less informative bounding-box images (low-quality or occlusions). With this approach and setting, an additional 2.0% Rank-1 performance increase was obtained for our final re-ID stage.

### 6.7. Travel-Time Selection

In this function, the added value of database filtering was evaluated, based on the implied travel time (see Section 3.2, G). This function can be directly applied to other object classes as well, such as persons or vehicles [25].

Inspection of our Vessel-reID dataset revealed that the vessel travel time between the two cameras was roughly 32 min. Additionally, we found that from all vessels in our dataset, only two vessels required more than 2 h to pass. Hence, we excluded any possible matches from the gallery where the implied travel time was longer than 2 h. The travel time was determined by comparing the time difference between the vessel occurrence time and the appearance times of the same vessel in the gallery tracklets. This function was applied on top of our previously discussed approach to concentrate on the largest bounding-box images in the query tracklet. In Table 4, an incremental overview of each function is presented in our final re-ID stage, showing its individual influence on re-ID performance. The results highlight that including travel-time information significantly improved re-ID performance, by 2.2% Rank-1 and 3.8% mAP for MGN.

### 6.8. Only Cross-Camera Samples in Gallery

In the functions discussed above, we improved the Default Querying Approach but still adopted the common database practice for comparison purposes with re-ID literature. Here, as mentioned in Section 3.2, each query was matched to a combined gallery, consisting of all images from all cameras. The only images that were excluded from the gallery were the images from the query tracklet, i.e., those from the same class (e.g., vessel) that also originate from the same camera as the query. Using the combined gallery of all cameras, it was more likely that mismatches could occur. However, given our application, we specifically required that a query from one camera was matched to the gallery images of another camera. In this subsection, we defined a function that changed the previous generic matching procedure such that it only considered images of the other camera in the gallery, thereby restricting ourselves to only cross-camera matching. An evaluation with this constraint provided insight into how the vessel-speed enforcement application would ultimately perform in practice.

For this application, we obtained the final re-ID performance by applying this cross-camera matching function on top of all previously introduced refinement functions. That is, by applying the proposed only cross-camera matching in combination with the Top-25 Tracklet-based Querying Approach, where only the largest bounding-box images per tracklet were considered and additional travel-time selection was included. As presented in the bottom row of Table 4, a re-ID performance score of 88.9% Rank-1 and 83.5% mAP was achieved ultimately for MGN.

## 7. Discussion

An important point of discussion is the fact that the system design is actually based on a closed-set gallery, which refers to the fact that there is always a matching vessel for each query image in the gallery. This effectively means that in our experiments we considered a closed-set re-ID problem. However, in an actual real-world application of our system we have to consider re-ID as an open-set problem, because for some vessels there are no matches available. Images of a newly appearing vessel (sailing into the trajectory covered by the cameras) should not/cannot result in a matching historic vessel. Moreover, vessels can be removed already actively from the gallery set after their maximum travel time has passed, since we are only interested in speeding vessels. This effectively reduces the gallery set and removes the closed-set constraint. Although in reality there may not be a true match in the gallery set, the proposed system always returns the best-matching vessel. This emphasizes the value of the Rank-1 evaluation score. Therefore, the system design needs to be adjusted, such that also a non-match can be detected, e.g., by applying a threshold on the matching score. The impact of this modification is left for future work.

## 8. Conclusions

In this paper we have proposed a first camera-based vessel-speed enforcement system. This system uses a setup with two cameras, spaced several kilometers apart. To our knowledge, this is the first time that an automated trajectory speed measurement system for vessels has been investigated and reported. The system performs vessel detection and tracking per camera view and employs a re-ID system that links vessels between the two cameras. Multiple images are collected for each vessel and stored in a gallery database for visual re-ID. Newly detected vessels in one camera (query) are compared to the gallery set of all vessels detected by the other camera.

Novel dataset. We have introduced the novel Vessel-reID dataset. This extensive vessel dataset was captured by two camera positions mounted 6 kilometers apart at a canal in the Netherlands. During four days, a total of 2474 different vessels were captured. Each vessel is represented by multiple images captured along its trajectory in the camera view, resulting in 18,338 camera images (video frames), totalling to 136,888 vessel image crops (bounding boxes). The set contains a large variation in vessel appearance, varying from large commercial barges to sailing ships and small dinghies. Most vessels in the set are pleasure crafts. The total set has been separated in train and test sets by using separate days of data capture, resulting in a test set with different light/weather conditions, instead of mixing data of all captured conditions in the test and train sets, which is common in re-ID literature.

Vessel detection. We have investigated the effect of the dataset on the detection performance. It was found that the SeaShips-7000, VCA-VCO and our Vessel-reID dataset are providing different performances and training with all datasets jointly results in the best performance. The performance of various CNN architectures for detection has been evaluated. The Faster-RCNN (85.0% R@95P) and SSD512 (84.0% R@95P) detectors both result in a high detection accuracy, whereas FCOS (78.0 R@95P) and YOLO-v3 (71.0% R@95P) obtain a lower performance. When visually inspecting detection results, it seems that Faster-RCNN generalizes better to the large variation of vessels and can handle occlusions better. All detection models are able to accurately detect ships of different sizes, but show a small performance decrease for small vessels. We have measured the computational complexity of all detectors and found that SSD512 yields optimal performance with respect to accuracy (20.1 frames per second). For the re-ID of vessels, we have experimentally validated that for most vessels, a large portion of the total trajectory is detected by the SSD detector. All vessel types are detected, with an exception of a few very small rowboats having a limited size of only a few pixels. Summarizing, the detection system accurately recognizes vessels, which enables further application of vessel re-identification.

Re-identification. We have performed an extensive comparison of two popular re-ID algorithms: TriNet and the Multiple Granularity Network (MGN), where MGN consistently outperforms TriNet in all of our experiments. Directly applying these algorithms with the common evaluation method in which single images of vessels are compared for re-ID, results in a (baseline) score of 55.9% Rank-1 (49.7% mAP) for TriNet, while MGN obtains 68.9% Rank-1 (62.6% mAP). Our main contribution for re-ID is in applying matching with multiple images from the trajectory of a single query vessel, in contrast to matching based on a single image only. This aspect alone significantly increases the re-ID performance with 5.6% Rank-1 (5.7% mAP) for MGN. When additionally emphasizing the largest images in the query tracklet that contain more fine details of the query vessel, another 2.0% Rank-1 (1.4% mAP) is added to the re-ID performance. Next, we have proposed application-specific filtering of matches to further improve re-ID performance. First, applying travel-time selection adds 2.2% Rank-1 (3.8% mAP), leading to a combined performance of 78.7% Rank-1 and 73.5% mAP, without re-ranking. Second, by further restricting to cross-camera matching only, we obtain the highest re-ID performance where 88.9% (Rank-1) of the vessels are correctly re-identified in the other camera (83.5% mAP). The proposed extra functions result in a significant increase in performance of +20% Rank-1 and +21% mAP, compared to the single-image baseline matching. The final re-ID contribution contains also a execution-time analysis of the proposed re-ID stage. A computational speedup is achieved by applying linear subsampling to all trajectories (query and database) to limit the number of images per trajectory. Extensive validation has shown that 20 images per trajectory yields the best balance between re-ID accuracy and execution time, resulting in a speedup of 4× (from 1250 to 275 msec per query) without loss in accuracy for the Rank-1 score and only a 0.3% mAP loss. With this setting, applying the proposed matching and travel-time selection procedure yields an average throughput of 3.6 vessels per second, which is clearly sufficient for the law-enforcement application, as vessels typically do not move fast.

Summarizing the final conclusions, the vessel recognition system efficiently detects all vessels that pass the camera view. Re-identification links unique vessels between the two cameras and achieves a high accuracy of close to 90% Rank-1 score. The proposed system enables the automated speed measurement of vessels over a large-distance trajectory. The proposed system can achieve real-time operation with 20.1-fps object detection and 3.6-vessels/s re-identification (where 0.05 vessels/s have been observed during peak hours). Although the current process of law enforcement still requires the intervention of a human operator, the performance of our automated system is very high and can be integrated to directly support law enforcement. Consequently, the system automatically captures 90% of all speeding vessels, while only leaving the operator a manually filtering task for the remaining incorrect 10%. All proposed improvements have been experimentally validated for the vessel class, but are expected to give similar re-ID performance gains for other classes such as persons.

## Figures and Tables

**Figure 1 sensors-21-04659-f001:**
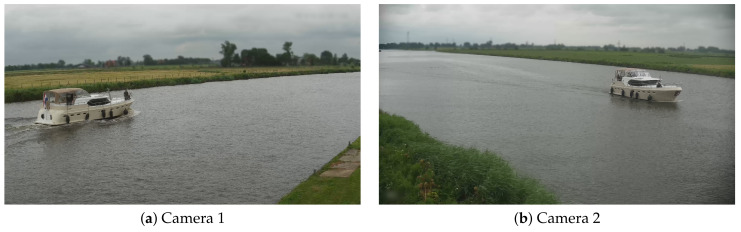
Example images of the same vessel appearing in Camera 1 (**a**) and 2 (**b**), spaced 6 km apart.

**Figure 2 sensors-21-04659-f002:**
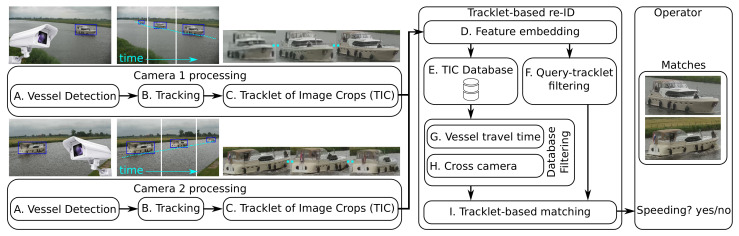
System overview, depicting the procedure at test time.

**Figure 3 sensors-21-04659-f003:**
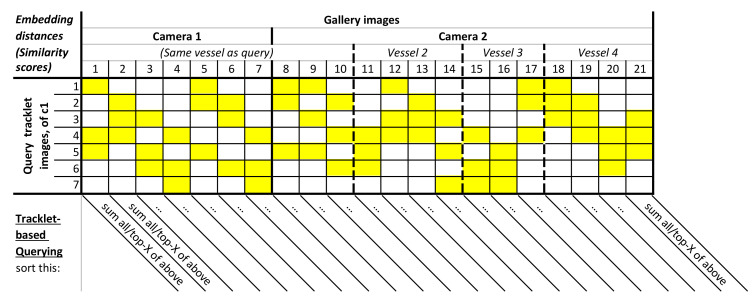
Visualization of our novel Tracklet-based Querying Approach. The yellow boxes indicate the top-3 query tracklet images per database-image (i.e., per column). For conciseness, there is only a single vessel visualized in the database of Camera 1 (c1).

**Figure 4 sensors-21-04659-f004:**
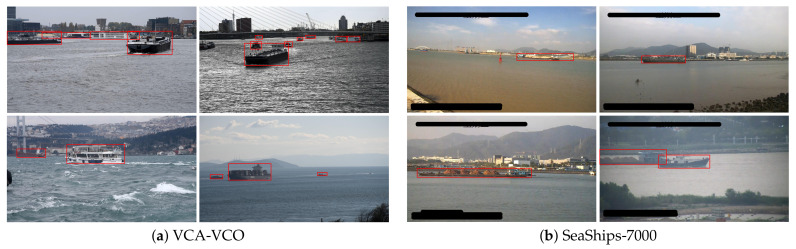
Example images from the SeaShips-7000 and VCA-VCO datasets with box annotations (red).

**Figure 5 sensors-21-04659-f005:**
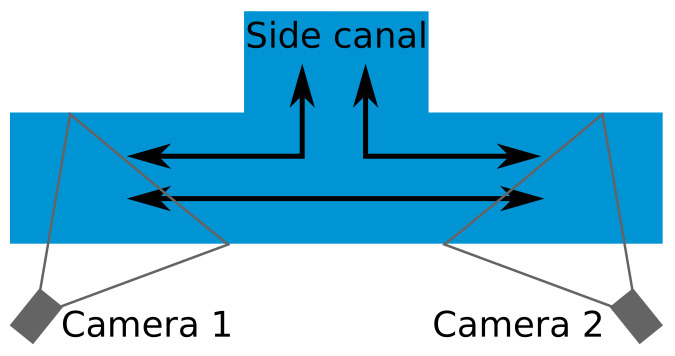
Schematic overview of the two cameras positioned along the main waterway (top-view).

**Figure 6 sensors-21-04659-f006:**
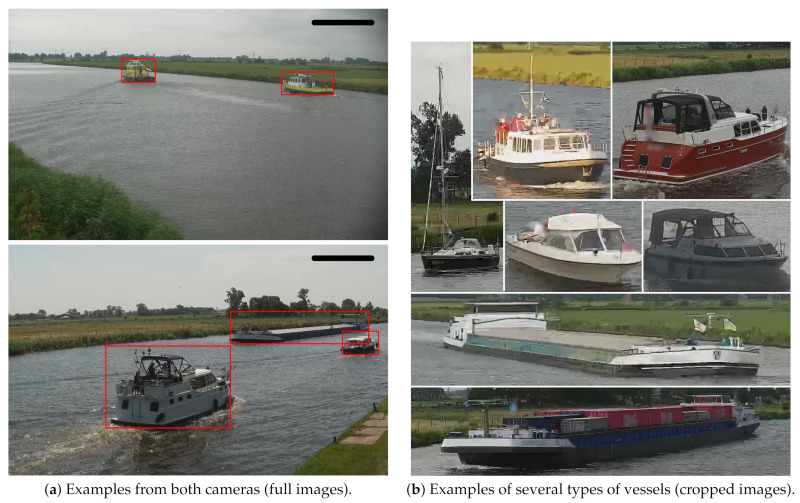
Example images from our Vessel-reID dataset.

**Figure 7 sensors-21-04659-f007:**
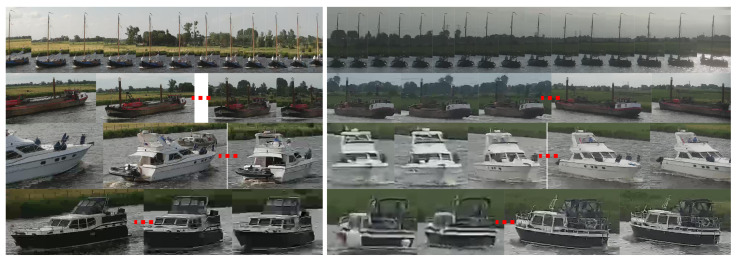
Example images of vessel trajectories in both Camera 1 (**left**) and Camera 2 (**right**). Each row shows a unique vessel trajectory in our dataset. Some intermediate images are skipped (red dots).

**Figure 8 sensors-21-04659-f008:**
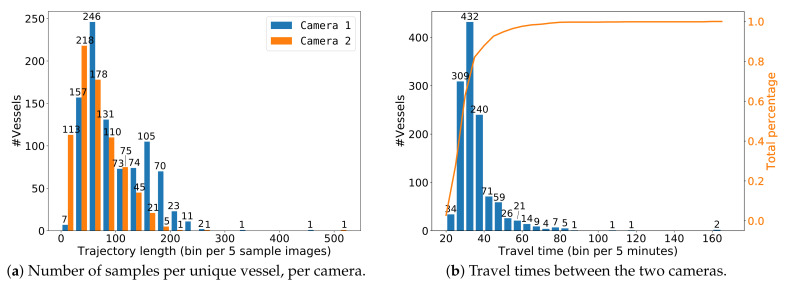
Statistics of the Vessel-ReID dataset, in terms of trajectory lengths and travel times.

**Figure 9 sensors-21-04659-f009:**
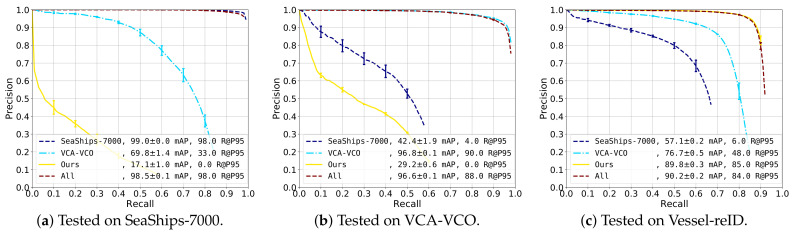
Detection performance: cross-validation, training SSD on different datasets individually as indicated in the caption on which it also is tested.

**Figure 10 sensors-21-04659-f010:**
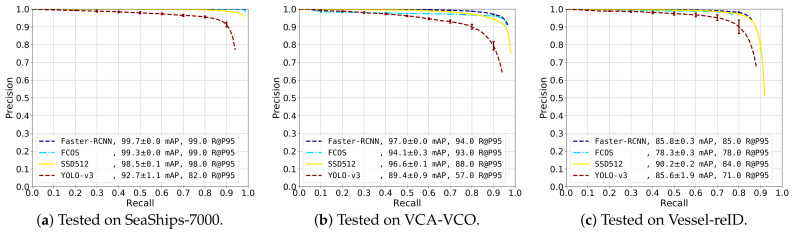
Detection performance: cross-validation of the selected detectors on different datasets, trained on all datasets combined and tested with the individual test sets per dataset.

**Figure 11 sensors-21-04659-f011:**
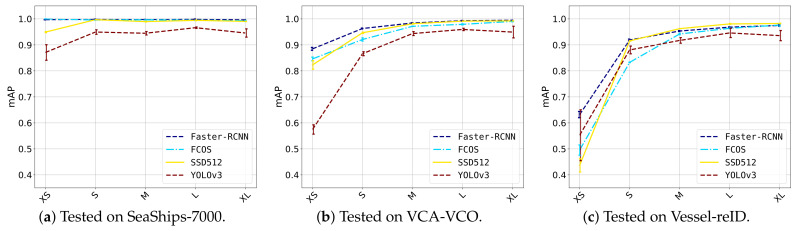
Detection performance: influence of vessel size for different detectors on the separate datasets.

**Figure 12 sensors-21-04659-f012:**
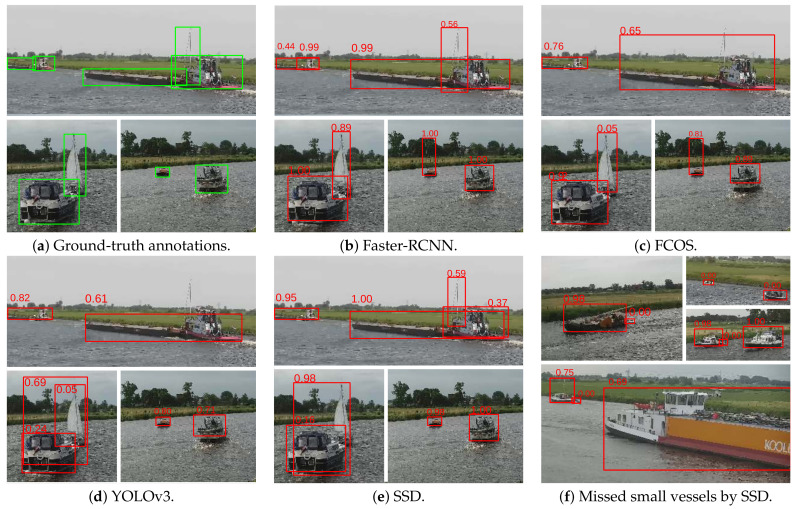
Examples of bounding-box detections and confidence scores generated by Faster-RCNN, FCOS, YOLO and SSD512 on the Vessel-reID dataset.

**Figure 13 sensors-21-04659-f013:**
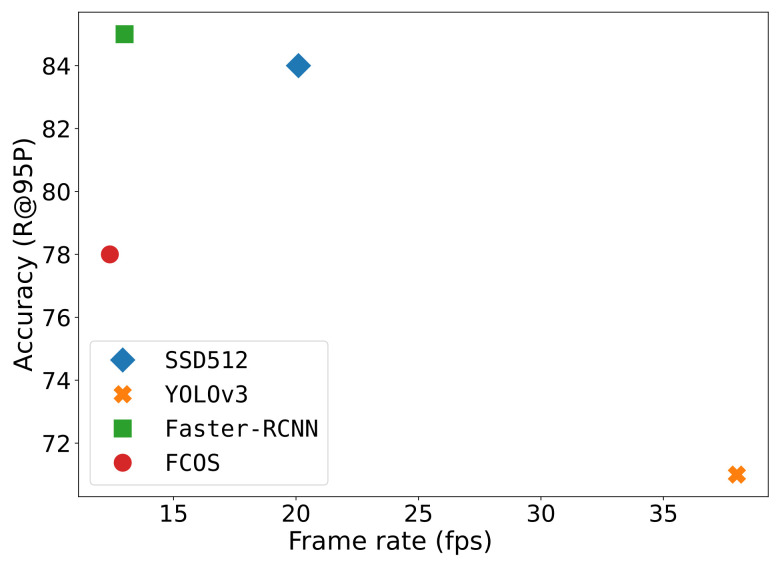
Trade-off between throughput speed and accuracy for the four detectors.

**Figure 14 sensors-21-04659-f014:**
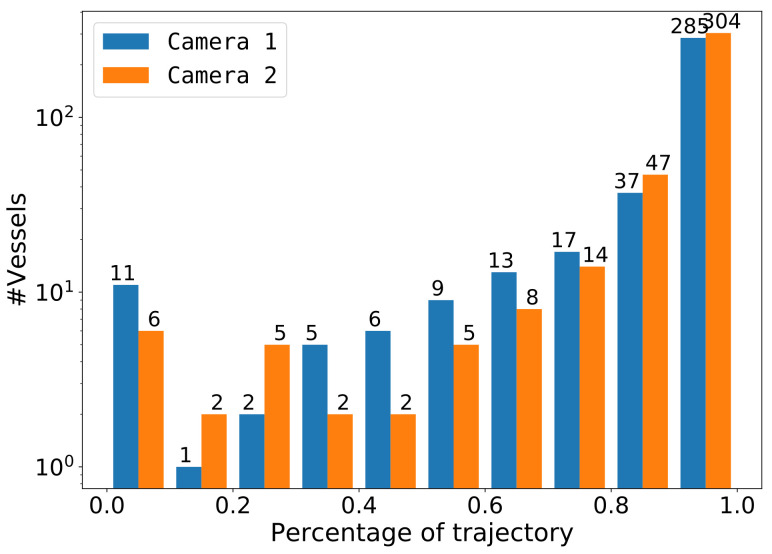
The number of vessels detected per camera for a given percentage of its actual trajectory.

**Figure 15 sensors-21-04659-f015:**
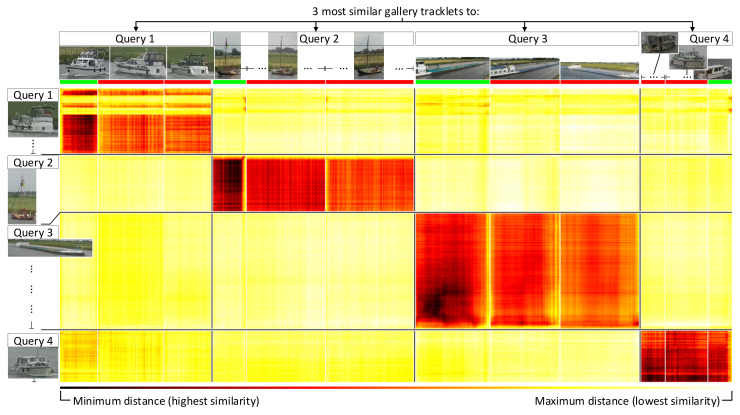
Distance matrix visualization of our Tracklet-based Query Approach for four query vessels (left side of the matrix). For each query, the distance values are shown from each image in its tracklet to each image in the depicted gallery tracklets (top side of the matrix). The thin horizontal red/green color bars underneath the gallery images indicate the ground-truth matching result with the indicated query (green was match, red mismatch). The colors in the matrix cells refer to the embedding distances between the query and the gallery images. These colors are scaled per query tracklet (row) for maximum contrast, where red hot means small distance and yellow white indicates large distance (see legend bar underneath). The query/gallery images at the left/top are only one visual example but represent the set of images in its query/gallery tracklet, where each individual color pixel in the matrix signals the individual matching distance by its color.

**Figure 16 sensors-21-04659-f016:**
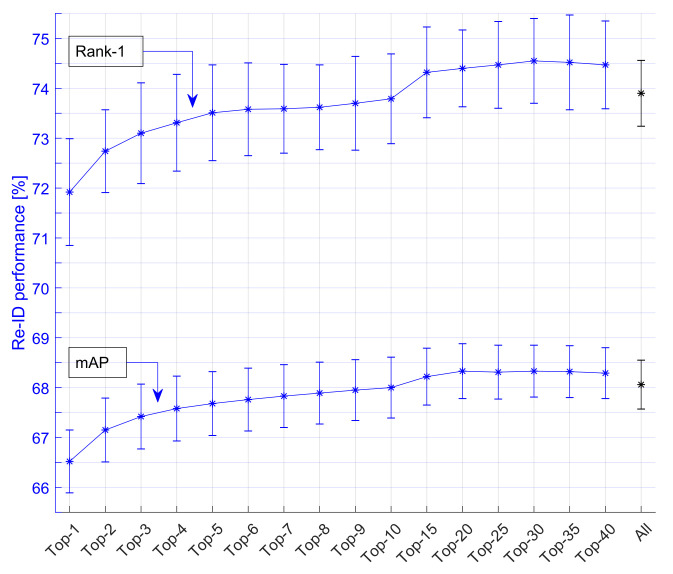
Re-ID performance expressed in Rank-1 (top curve) and mAP score (bottom curve), as a function of the number of top-matching images (*Top-X*) with the proposed Tracklet-based Query Approach. Results are reported as mean (stddev) over 10 MGN training cycles.

**Figure 17 sensors-21-04659-f017:**
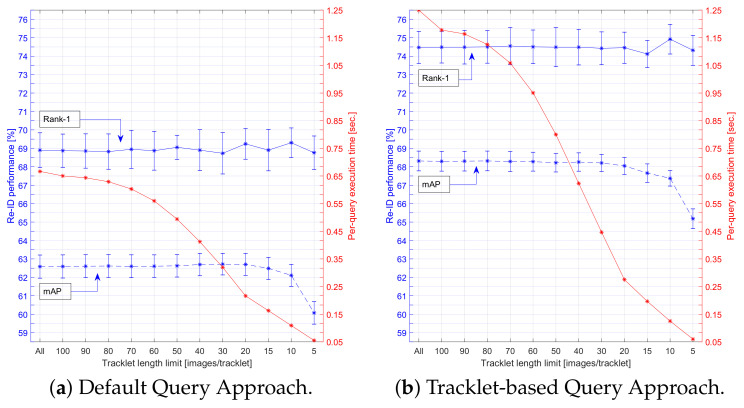
Re-ID performance expressed in Rank-1 (top blue curve) and mAP score (bottom blue curve), when applying linear subsampling to all tracklets (both gallery and query set) in order to improve computational efficiency (going rightwards). The color of the curves indicate which *y*-axis applies. Results are presented as mean (stddev) over 10 MGN cycles, and Tracklet-based matching (**b**) is based on Sum-of-top-25.

**Figure 18 sensors-21-04659-f018:**
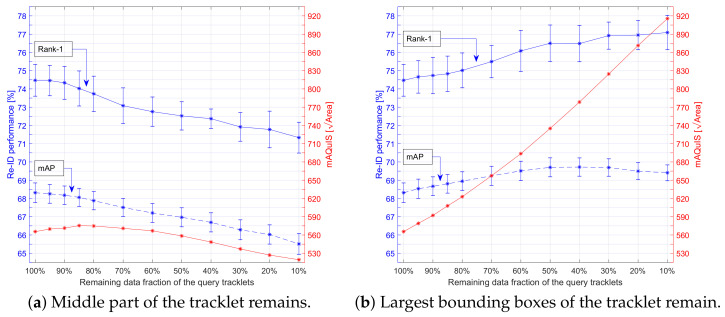
Accuracy of the re-ID performance (blue) as a function of the remaining data fraction of the query tracklets for our Tracklet-based Query Approach. The mean Average Query Image Size (mAQuIS) is shown in red, where image refers to bounding box. The curve color indicates which *y*-axis is applicable. Results reported as mean (stddev) over 10 cycles for MGN.

**Table 1 sensors-21-04659-t001:** Statistics of VCA-VCO, SeaShips-7000 and Vessel-reID (ours) datasets.

	Resolution	Images		Vessel Annotations (Unique)		Application	
Dataset	(Pixels)	Train	Test	Train	Test	Detection	ReID
VCA-VCO	1920 × 1080	10,000	1000	28,260 (N.A.)	2818 (N.A.)	✓	
SeaShips	1920 × 1080	31,455		≈500 (≈60)		✓	
SeaShips-7000	1920 × 1080	3500	3500	4,538 (N.A.)	4683 (N.A.)	✓	
Vessel-reID (ours)	3840 × 2160	81,984	28,044	577,794 (2226)	243,534 (787)	✓	✓

**Table 2 sensors-21-04659-t002:** Number of vessels per day moving in each direction.

Direction		Day 1	2	3	4
From	To				
Cam1	Cam2	178	104	167	181
Cam2	Cam1	155	134	176	142
Cam1	Side-canal	34	23	45	31
Cam2	Side-canal	31	30	50	33
Side-canal	Cam1	25	28	41	33
Side-canal	Cam2	24	27	40	44
**Total TICs Cam1**		392	289	429	387
**Total TICs Cam2**		388	295	433	400
**Dataset split**			**Train**	**Test**	

**Table 3 sensors-21-04659-t003:** Upper and lower pixel area limits for the different bins of each dataset used for evaluation.

Bin Limits	SeaShips-7000	VCA-VCO	Vessel-reID
Resolution [px]	1920×1080	1920×1080	3840×2160
Smallest	608	276	255
10% area	11,232	1760	4918
30% area	36,432	4880	15,790
70% area	149,490	24,332	91,917
90% area	294,867	89,518	376,913
Largest	813,656	766,080	4,919,722

**Table 4 sensors-21-04659-t004:** Re-ID performance on our Vessel-reID dataset, when incrementally applying the novel functions on tracklet-based querying, remaining data selection, travel time, and cross-camera samples (see Section 3.2). Bold numbers indicate the reference values for both the ’Real-time optimized’ and ’Largest BB images only’ results. Results reported as mean (stddev) over 10 training cycles.

	TriNet		MGN	
Description	Rank-1	mAP	Rank-1	mAP
Vessel-reID (baseline)	55.9(±1.4)	49.7(±1.0)	68.9(±0.9)	62.6(±0.6)
+ Tracklet-based Querying				
All	61.9(±1.4)	55.0(±1.0)	73.9(±0.7)	68.1(±0.5)
Top-10	63.7(±1.1)	56.3(±0.8)	73.8(±0.9)	68.0(±0.6)
Top-15	64.0(±1.2)	56.5(±0.9)	74.3(±0.9)	68.2(±0.6)
Top-25	63.8(±1.2)	56.4(±0.9)	74.5(±0.9)	68.3(±0.5)
+ *Real-time optimized*	62.2(±1.4)	55.4(±1.0)	74.5(±0.8)	68.0(±0.5)
+ Largest BB images only	66.3(±1.2)	58.5(±1.0)	76.5(±1.0)	69.7(±0.5)
+ Travel-time selection	70.4(±1.2)	64.1(±0.8)	78.7(±1.0)	73.5(±0.4)
+ Only cross-camera	80.1(±0.9)	74.1(±0.6)	88.9(±0.6)	83.5(±0.2)
+ *Real-time optimized*	79.6(±0.8)	73.9(±0.6)	89.3(±0.5)	83.3(±0.2)

## Data Availability

Not applicable.

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
