# Peer review of "Multi-Camera Vessel-Speed Enforcement by Enhancing Detection and Re-Identification Techniques â€"

_sensors, 2021, doi:10.3390/s21144659_

Round 1
Reviewer 1 Report
Proposed article is interesting and important. There is some general questions and/or unclear issues.
1) Please explain why the Authors decided to place the cameras observing the moving vessels in such a way that the movement in opposite directions is observed? From the point of view of the efficiency e.g. of the reID algorithm, would it not be more useful to observe vessels from a similar, preferably the same, perspective?
2) The title of the article includes the phrase "Vessel-Speed Enforcement" which suggests that the one of the describe main research problem will be determining the speed of vessels. The article lacks an in-depth analysis of this issue. Please complete the article with the scope related to determining the vessels' speed and indicate the methods of its validation and verification.
Reviewer 2 Report
===== Synopsis:
The study introduces a system estimating the speed of vessels passing through a canal by means of two cameras separated by some distance (several hundred meters). A novel vessel dataset is introduced and a variety of networks are tested. The study is more concerned with Software evaluation than with Hardware evaluation.
===== General Comments:
The study reads well in general, and I enjoyed working my way through, but it is suffocating and confusing nevertheless toward the end. The reasoning of the author's choice for DeepNetworks is good - it's hard to comprehend the chaos of networks in general, but the authors introduced that part very clearly. I've learned something from that.
The study could be written up a bit shorter. Some of the experiences that the authors have made, i.e. optimal parameter ranges, can be introduced as "a value of x appeared optimal". This is not to say that those experiences are interesting, but I feel they rather belong into a PhD thesis (or similar). Some aspects appear irrelevant, e.g. the processing duration for detection and tracking - the author's even conclude that they are of lesser importance. More suggestions where to shorten follow further below.
Other comments:
The two cameras appear to point in opposite directions. Camera 1 takes the frontal view of the vessel, camera 2 takes the rear view. That means the Re-ID process matches the rear view to the frontal view somehow? Or is the vessel also seen partially from the side in both cameras?, in which case the Re-ID is much easier.
Speed estimation: is this done twice? Once for one camera view using tracking, and once by taking the time traveled using Re-ID in the second camera?
Tracking: what exactly is it useful for (apart from speed estimation in one viewpoint)? Taking single shots - without linking them to a track - is not enough for Re-ID? It appears to me that is what you conclude, but I am not quite certain.
5.4. Detection execution-time analysis: why is this so important (appear to me irrelevant). I would have assumed that to be important for fast vehicles, but not for vessels that travel generally < 25 km/h. For your tracklets you subsample frames anyway.
The conclusions read more like an extensive summary. A more compact version would be better.
===== Specific Comments:
- Line 447: "We conclude...": the conclusion isn't quite clear. You mean that if you use only a SINGLE dataset (and not all 3) that the detection accuracy is still good enough?
It appears to contradict with the conclusion starting line ca. 463.
- Paragraph starting with line 43: it wasn't immediately clear that you were talking about automobile traffic first. Then you switched to maritime traffic.
- Sentence explaining tracklet (line 317) should be introduced earlier. I had already wondered what exactly it is, but is explained late.
- If the manuscript needs to be shortened (ie. due to page limits), then I think the first few paragraphs explaining the need for this type of identification can be shortened to one paragraph. And Figure 4 showing example images of other datasets can be dropped. And section 5.4 (execution time analysis).
